# Copepods as Indicators of Different Water Masses during the Northeast Monsoon Prevailing Period in the Northeast Taiwan

**DOI:** 10.3390/biology11091357

**Published:** 2022-09-16

**Authors:** Yan-Guo Wang, Li-Chun Tseng, Rou-Xin Sun, Xiao-Yin Chen, Peng Xiang, Chun-Guang Wang, Bing-Peng Xing, Jiang-Shiou Hwang

**Affiliations:** 1Institute of Marine Biology, National Taiwan Ocean University, Keelung 202301, Taiwan; 2Third Institute of Oceanography, Ministry of Natural Resources, Xiamen 361005, China; 3Center of Excellence for Ocean Engineering, National Taiwan Ocean University, Keelung 202301, Taiwan; 4Center of Excellence for the Oceans, National Taiwan Ocean University, Keelung 202301, Taiwan

**Keywords:** copepods, indicator species, Kuroshio Current, Northwest Pacific Ocean

## Abstract

**Simple Summary:**

How the origin and pathways of water masses can be traced by particular bioindicators remains an intriguing issue in biological oceanography. In the present zooplankton study focusing on copepods, calanoid copepodites were most abundant, with an average abundance of 774.24 ± 289.42 (inds. m^−3^) in the northeastern waters of Taiwan during the prevailing northeast monsoon, followed by the dominant copepod species *Paracalanus aculeatus* and *Clausocalanus furcatus*. According to hydrological parameters, the water masses were mainly derived from northeast monsoon surface waters, Kuroshio intrusion water, and mixed water masses. Indicator species were *Temora turbinata*, *Calanopia elliptica*, and *Canthocalanus pauper* in the northeast monsoon-derived water mass. *Farranula concinna* and *Copilia mirabilis* represented suitable indicators for the Kuroshio intrusion water mass in the research area. In the mixed water mass, the indicator species were *Paracandacia truncata*, *Oncaea clevei*, and *P. aculeatus* in the research area during the sampling campaign in late autumn.

**Abstract:**

During this research, the average surface temperature, salinity, dissolved oxygen, and pH were 24.65 ± 1.53 (°C), 34.21 ± 0.07 (PSU), 6.85 ± 0.18 (mg/L), and 8.36 ± 0.03, respectively. Based on these environmental parameters, stations were arranged into three groups. Group A represents stations located around Keelung Island with the relative highest average dissolved oxygen, lowest average temperature, and pH values. Instead, the lowest average dissolved oxygen and highest average temperature, salinity, and pH values were recorded at the offshore stations. Keelung Island area was charged by cold water masses, which were driven by the Northeast monsoon, and stations in group C were affected by the Kuroshio Current. Kueishan Island area was mainly affected by mixed water masses resulting from the Kuroshio intrusion and monsoon-derived cold water. In this study, a total of 108 copepod species were identified, with an average abundance of 774.24 ± 289.42 (inds. m^−3^). Most species belong to the orders Calanoida and Poecilostomatoida, with an average relative abundance (RA) of 62.96% and 30.56%, respectively. Calanoid copepodites were the most dominant group, with a RA of 28.06%. This was followed by *Paracalanus aculeatus,* with a RA of 18.44%. The RA of *Clausocalanus furcatus* and *Canthocalanus pauper* was 4.80% and 3.59%, respectively. The dominant species *P. aculeatus*, *C. pauper*, *Paracalanus parvus,* and *Temora turbinata* were positively correlated with dissolved oxygen and negatively correlated with temperature in the surface waters. pH showed a negative correlation with *P. parvus* and *T. turbinata*, while the temperature was negatively correlated with these two dominant species. Indicator species were selected by an indicator value higher than 50%. *Temora turbinata*, *Calanopia elliptica*, *C. pauper*, *Euchaeta concinna*, *Temora discaudata*, *Acartia pacifica*, *Macrosetella gracilis*, *Corycaeus speciosus,* and *P. parvus* were considered as monsoonal cold water indicator species in Group A. Indicator copepod species for the Kuroshio Current were *Farranula concinna*, *Copilia mirabilis*, *Candacia aethiopica*, *Corycaeus agilis*, *Farranula gibbula* and *Acrocalanus monachus* in the study area. *Paracandacia truncata*, *Oncaea clevei*, *P. aculeatus,* and *Centropages furcatus* were considered suitable indicators for mixed water masses.

## 1. Introduction

In the marine environment, zooplankter is exposed to and sensitive to physical, chemical, and biological factors [1,2,3,4]. Their population dynamics were shown to respond closely to seasonal temperature variations [5,6,7]. Copepods provide the most abundant zooplankton assemblages in marine ecosystems [8,9,10]. In many places, they account for the majority of mesozooplankton abundance and could comprise about 70% or more of the zooplankton community in abundance and about 30% of its biomass [11]. Copepods play a key role in aquatic food webs and biogeochemical cycles by transferring nutrients and energy from primary producers to higher trophic levels [12,13,14,15,16]. As they are highly dependent on environmental conditions (e.g., chemical and organic contaminants, temperature, and salinity), being able to give a fast response at the ecologically relevant population and community levels, copepod species have long been used as bioindicators of environmental quality and water mass origin [17,18,19,20]. Marine copepods are an excellent tool for evaluating the impact of marine pollution along coastal regions because they quickly respond to different types of stressors in different ways (e.g., decreased fecundity, gender bias, molecular responses, mortality) [21,22,23,24,25,26]. Marine copepods can also track changes in environmental conditions that occur at interannual to multidecadal scales [27,28,29,30,31].

Eddies and coastal currents could directly promote the migration of zooplankton by physical transport or modify water temperature, salinity, Chl *a* concentrations, dissolved oxygen (DO), and other seawater environmental variables which indirectly control the distribution of zooplankton [32,33,34,35,36,37]. Previous studies reported that both the Kuroshio onshore intrusion and the southwestward current along the slope frequently prevail in the northeast of Taiwan [38,39]. Due to local monsoons, heat flux, and mesoscale eddies east of Taiwan, the Kuroshio Current showed significant seaward migration in summer and on-shelf encroachment and intrusion in winter [40,41,42,43]. There was a year-round upwelling via the subsurface Kuroshio water onshore intrusion in the northeast of Taiwan [44,45,46]. Upwelling was shown as an important pathway for shelf-ocean exchanges of dissolved and particulate material between the East China Sea (ECS) and the Kuroshio Current, which commonly causes a higher phytoplankton production [47]. The well-mixed intrusion of Kuroshio water and the continental shelf current, plus the upwelling, critically affects the composition and abundance of oceanic biota around the island of Taiwan [35,48,49,50,51,52]. Previous studies indicate a correlation between the abundance of biota and the water masses in northeast Taiwan [53]. Copepods also show a significant diel vertical migration in the upwelling area of northeastern Taiwan [54]. Previous studies indicated that seasonal successions of dominant copepod species range from 25.0% to 61.9% in northeast Taiwan, with, among others, the following dominant species *Cosmocalanus darwinii*, *Canthocalanus pauper*, *Clausocalanus furcatus*, *Paracalanus aculeatus* and *Temora trubinata* [8,12,52,53,55]. The dominant species, *T. turbinata,* shows a significant negative correlation with surface water temperature and seasonal variation in the research area [51,56].

Previous studies indicate that the planktonic copepods in the waters of northeastern Taiwan are affected by the interactions between East China Sea waters and Kuroshio waters [8,12,51,52,53,55,56]. The hydrological characteristics of the waters in this region change quite rapidly. To date, the relationship between the biogeographic distribution of planktonic copepods and the water masses in this region during the prevailing northeast monsoon period is unclear. Therefore, this study aims at testing two hypotheses: (1) when the northeast monsoon prevails, the waters of northeastern Taiwan are affected by the interaction between East China Sea waters and Kuroshio waters, resulting in differences in the geographic distribution of planktonic copepods; (2) changes in temperature and salinity affect the occurrence and abundance changes of dominant copepod species. The aim of the present study was to understand the potential effects of environmental factors on copepod communities and characteristics during the prevailing northeast monsoon period in autumn in northeast Taiwan. For this purpose, changes in the plankton copepod species composition, dominant species, and community characteristics were analyzed here. Indicator values were calculated based on species distribution and abundance to demarcate different water masses in the research area.

## 2. Materials and Methods

Samples were collected in the northeast of Taiwan (Figure 1) during the prevailing northeast monsoon period from 1st to 3rd November 2019 by research vessel II. At each sampling station, zooplankton was collected using a west pacific standard zooplankton net with a mouth diameter of 45 cm and a mesh size of 200 μm. The net was horizontally towed from the surface to the 5 m depth layer with a speed of about 1.0 m/s. A flow meter was mounted in the center of the net mouth to calculate the volume of filtered water. The net was fitted with a hard cod end to ensure that the collected organisms were in good condition. Samples were immediately preserved in 4% seawater buffered formaldehyde solution. Water temperature, salinity, dissolved oxygen, and pH were measured by sensors that were mounted on a SeaBird rosette sampler.

The images of the High-Resolution Sea Surface Temperature (SST) were obtained from the Fisheries Research Institute, Council of Agriculture, Taiwan. The images of the sea surface temperature were used to indicate the range of different water masses in the research area.

In the laboratory, zooplankton samples were split with a Folsom plankton sample splitter until subsamples contained not less than 200 individuals, and subsamples were enumerated. Copepods were identified to species level where possible and counted under a stereomicroscope (Leica MZ95). Several taxonomy studies were referred to when identifying copepod species [57,58,59,60]. Abundance was provided as the number of individuals per cubic meter (ind/m^3^).

Regional differences in mean abundance among groups were tested by one-way analysis of variance (ANOVA), and Tukey’s honestly post hoc test was applied to distinguish which groups were statistically different in their community structure among the sampling zones using the statistical software package SPSS v24. Dominant species compositions were identified by the decorana function in Vegan package, with the axis length approximating 1.39, which was less than 3. This way, a redundancy analysis (RDA) was performed to correlate environmental parameters with dominant species compositions and distribution using vegan package version 2.5–7, and the RDA diagram was drawn by plot2 package version 3.3.5 in R language [61]. Indicator species are species that are sensitive to change or whose abundance in a given area indicates certain changes in environmental conditions [62,63]. They are used to monitor these changes for fundamental research and to provide information for effective environmental management. In order to find potential indicator species in the groups which resulted from the cluster analysis, the program Indicator Value (IndVal) was applied [64]. Species were defined as indicator species with IndVal higher than 50%. IndVal was calculated as:(1)$${\mathrm{IndVal}}_{ij}=A_{ij}*B_{ij}*100$$
where $A_{ij}=N_{individuals_{ij}}/N_{individuals_{i}}$, and $B_{ij}=N_{site_{ij}}/N_{site_{j}}$.${\;A}_{ij}$ was a measure of site specificity, where $N_{individuals_{ij}}$ was the mean number of *i* species across sites of group *j*, and $N_{individuals_{i}}$ was the sum of the mean numbers of individuals of species *i* over all groups. $B_{ij}$ was a measure of group fidelity, where $N_{site_{ij}}$ was the number of sites in group *j* where species *i* was present, while $N_{site_{j}}$ was the total number of sites in that group [64].

## 3. Results

### 3.1. Spatial Changes in Hydrographic Conditions

During this research, the northeast monsoon prevailed, and the southwest monsoon gradually weakened in the Northwest Pacific Ocean. The northeast monsoon drives the cold surface water southwestward to the East China Sea (Figure 2A). The cold water mass encountered the warm water mass, which originated from the Kuroshio onshore intrusion in the northeast of Taiwan. Research results showed that the salinity variations among sampling stations were not significantly different (Figure 2C). However, the surface temperature showed a more drastic variation than salinity (Figure 2B). Most stations in the east offshore area had the highest temperatures indicating that this area was mainly charged by Kuroshio intrusion water. Stations around Keelung Island showed a relatively low temperature caused by the monsoon-derived cold water intrusion. The water properties around Kueishan Island were relatively moderate, indicating a good mix of Kuroshio intrusion and monsoon-derived water masses in this area.

As Figure 2B indicated, the surface water temperature was highest offshore. The lowest temperature was recorded around Keelung Island station. However, stations around Kueishan Island had surface water temperatures ranging from 22.04 to 27.39 (°C) with an average of 24.65 ± 1.53 (°C). Surface water salinity showed uniform characteristics and was without drastic changes between stations (Figure 2C). The average surface salinity was 34.21 ± 0.07 (PSU), with the highest at 34.32 (PSU) in the offshore S4 station and the lowest at 34.01 (PSU) at the A17 station in the Keelung Island area. Surface water dissolved oxygen showed a negative correlation with surface temperature (Figure 2D). High levels of dissolved oxygen were distributed in the western coastal area with relatively low temperatures. In the eastern offshore area, the surface temperature was higher, and the dissolved oxygen was relatively lower. The average dissolved oxygen was 6.85 ± 0.18 (mg/L), with the highest at 7.16 (mg/L) at the A15 station and the lowest at 6.53 (mg/L) at the A4 station during this study. The surface pH value variations showed a positive correlation with surface temperature (Figure 2E). The highest pH value was recorded in the offshore area, and the lowest pH value appeared around Keelung Island. The pH value showed a drastic variation around Kueishan Island. The surface water average pH value was 8.36 ± 0.03, with the lowest value at 8.31, being recorded at A17 station in the western coastal area, and the highest value at 8.40 at A5 and A6 stations in the eastern area and S4-1 and S5 stations near Kueishan Island.

The hydrographic parameters were calculated according to their Bray-Curtis similarities at different research stations, and the complete linkage was used for k means cluster analysis. The research stations were separated into three groups based on cluster analysis (Figure 3). Stations that were located around Keelung Island in the west of the research area were arranged into group A with a similarity of 98.81% with a Pi value of 32.66 at a 0.1% significance level. The lowest average surface temperature was 22.78 ± 0.58 (°C), and the highest average dissolved oxygen with 7.07 ± 0.08 (mg/L) was found in group A. Average salinity and pH were 34.19 ± 0.10 (PSU) and 8.33 ± 0.03, respectively, in group A. Stations around Kueishan Island and A12 fall into group B, with a similarity of 99.14% and a Pi value 32.78 at 0.1% significance level, with a median average surface water temperature of 24.31 ± 0.37 (°C). The average salinity, dissolved oxygen, and pH were 34.17 ± 0.03 (PSU), 6.89 ± 0.04 (mg/L), and 8.36 ± 0.02, respectively, in group B. The remaining stations, being located in relatively deeper waters in the east of the research area, were clustering to group C. Similarity was 98.63%, with a Pi value of 32.87 at a 0.1% significance level. Stations in this group represented oceanic waters with relatively higher average surface temperature and salinity. The average surface temperature was highest in this group at 26.52 ± 0.63 (°C). The lowest dissolved oxygen was also recorded in this group, with 6.63 ± 0.07 (mg/L). Average salinity and pH were 34.27 ± 0.04 (PSU) and 8.39 ± 0.01 in group C.

### 3.2. Species Composition and Spatial Characteristics of Copepod Communities

In total, 108 species were identified to species level, and 13 taxa were identified as copepodites. All species belonged to Calanoida, Cyclopoida, Harpacticoida, and Poecilostomatoida. As Figure 4 indicated, either in terms of species number or average abundance, calanoid species were most dominant; the average abundance of calanoids was 612.95 ± 296.27 (inds. m^−3^), with a total average abundance of 79.17% and 68 species accounting for 62.96% of the total species number. Poecilostomatoid species represented 30.56% of the total species number with an average abundance of 130.22 ± 59.81 (inds. m^−3^), which accounted for 16.82% of the total average abundance. Both the species number and average abundance of Cyclopoida and Harpacticoida accounted for a relatively low percentage. The species number of Cyclopoida and Harpacticoida provided 6.48% of the total species number. The average abundance of Cyclopoida and Harpacticoida were 24.31 ± 20.98 and 8.70 ± 8.15 (inds. m^−3^), respectively, with a total of 4.01% of the total average abundance altogether.

Figure 5 shows the horizontal distribution characteristics of the pelagic copepod assemblage in the research area. The abundance was higher in the western inshore area than in the eastern offshore area. The highest abundance was recorded at the A16 station with 1510.19 (inds. m^−3^). This was followed by 1228.38 (inds. m^−3^) at A12 station, which was located nearby Keelung Island. Both at A16 and A12 stations, *P. aculeatus* was the most dominant species, with a relative abundance of 36.85% and 34.17%, respectively. The RA of calanoid copepodites was 20.03% and 10.51% at stations A16 and A12, respectively. The lowest abundance was found at the offshore A4 station with 279.08 (inds. m^−3^). *Clausocalanus furcatus* was the dominant species at this station, with a RA of 24.59%. This was followed by calanoid copepodites and *Oithona plumifera,* with RA 14.75% and 8.20%, respectively.

### 3.3. Dominant Species Composition and Community Indicator Species

During the period of our present study, total copepod abundance ranged from 279.08 to 1510.19 (ind.m^−3^) in the area investigated here. Mean abundance (MA), average relative abundance (RA), and occurrence rate (OR) of dominant species in the research area are shown in Table 1. Calanoid copepodites were the most dominant, with MA and average RA being 217.24 (ind.m^−3^) and 28.06%, respectively. The OR and RA of *P. aculeatus* were 88.00% and 18.44%, respectively. It was followed by *C. furcatus* with an OR of 96.00% and an average RA of only 4.80%.

Based on a cluster analysis of environmental factors (Figure 3), pelagic copepods were compared in different environments. The highest average abundance was recorded in group A with 952.51 ± 278.29 (inds. m^−3^) and 71 species. This was followed by 838.59 ± 238.18 (inds. m^−3^) in group B, where 93 species were identified. The average abundance was only 552.07 ± 246.31 (inds. m^−3^) in group C with 75 overall species. The results of one-way ANOVA showed that the average abundance was significantly different between groups C and B (*p* = 0.02). The difference between groups A and C was also significant (*p* = 0.007). There was no obvious difference between groups A and B. Using IndVal, we found eight species that were significantly associated with Group C. In group B, the indicator species were *Paracandacia truncata*, *Oncaea clevei*, *P. aculeatus*, *Centropages furcatus* and copepodities of *Euchaeta* sp. Particular indicator species were *T. turbinat**a*, *Calanopia elliptica*, *C. pauper*, *Temora discaudata*, *Euchaeta concinna*, *Acartia pacifica*, *Macrosetella gracilis*, *Corycaeus speciosus* and *Paracalanus parvus* in group A. The mean abundance and IndVal of indicator species are shown in Table 2. The horizontal distribution of the indicator species showed an extreme correlation with water masses in the research area (Appendix A).

### 3.4. RDA Analysis of Dominant Species with Environmental Variables

Figure 6 shows the RDA ordinations of environmental factors with sampling sites and dominant species composition and distribution during this study. Stations of the offshore area were characterized by higher surface water temperature, salinity, and pH but with relatively lower dissolved oxygen. *Paracalanus aculeatus*, *T. turbinata,* and calanoid copepodites showed a positive correlation with dissolved oxygen and a negative correlation with surface water temperature, salinity, and pH during our research. *Oncaea media* and *F. gibbula* showed a slightly positive correlation with salinity.

## 4. Discussion

There is a year-round Taiwan Warm Current in the study area, which originates from the Taiwan Strait Current and Kuroshio intrusion across the slope of northeastern Taiwan [65,66]. Several studies also revealed that the current in the northeast of Taiwan was highly influenced by local wind forcing and the Ekman current induced by local wind forcing, which decayed substantially with increasing depth. The Ekman current was confined mainly to the upper 20 m, as confirmed by other studies [67,68,69]. During the northeast monsoon (September to April), northeasterly winds prevail, and the southward China Coastal Current brings cold surface waters from the Yellow Sea and the East China Sea into the northeast of Taiwan [70,71,72,73,74]. At the same time, the Kuroshio on-shelf intrusion was enhanced by the joint effect of baroclinicity and relief associated with winter cooling during the northeast monsoon period [33,40,75]. In the near-surface layer (0–2 m), the water temperature in I-Lan Bay was about 2 °C lower than the water temperature in the Kuroshio Current region [76]. Hsu et al. [76] proved that there was a significant thermal front outside I-Lan Bay, between the coastal or shelf waters from the northeast and Kuroshio Current based on data from drifters and shipboard acoustic Doppler current profiler information. Monthly moderate-resolution imaging spectroradiometer chlorophyll-*a* concentrations and attenuation coefficient at 490 nm data also proved the movement of water masses and ocean fronts and water masses in the research area [76]. During this research, the northeasterly wind prevailed; the southwestward cold water mass in the surface layer was strengthened with a surface water average temperature of 24.65 ± 1.53 (°C) in the research area. However, the average surface temperature of stations in the Kuroshio Current (KC) area was obviously higher than at Keelung Island and Kueishan Island. The chlorophyll-*a* concentration was 0.53 mg/m^3^ (winter) and 0.62 mg/m^3^ (autumn) in I-Lan Bay, but only 0.22 mg/m^3^ (winter) and 0.16 mg/m^3^ (autumn) in the KC area [76]. Higher chlorophyll-*a* concentration would expect more oxygen to be released by photosynthesis, which would lead to relatively higher dissolved oxygen levels in surface waters. In this study, the dissolved oxygen was obviously lower in the KC area than at Keelung Island and Kueishan Island. The average salinity and temperature were highest in the KC area compared to other areas in this study. Zhou et al. [77] reported that the KC is a salty, warm, and oligotrophic western boundary current of the North Pacific subtropical gyre.

In the marine environment, copepod abundance usually accounted for about 70% of the zooplankton communities [11]. Due to the short life span characteristics, the abundance of copepodites commonly accounts for 21–47% of the total zooplankton abundance and 59.77% to 90.82% of the total copepod abundance in field campaigns [78,79,80]. In the present study area, Wang et al. reported that the copepodites of the dominant species *T. turbinata* accounted for 27.29% to 60.30% of the total *Temora* abundance throughout the year [56]. In the present study, calanoid copepodites represented the most dominant group, which accounted for 28.06% of the total average copepod abundance with an OR of 100%. With the exception of calanoid copepodites, the dominant species composition coincided with previous research results in the research area [8,12,52,55].

Zooplankton distributions can be patchy due to multiple physical, chemical, and biological processes. Anandavelu [81] pointed out that the heterogeneous distribution of copepods along tropical coastal waters was attributed to the riverine influx and other environmental conditions. In the present research, surface water temperature, dissolved oxygen, and pH values were the main environmental factors that affected the spatial distribution of copepods. The heterogeneous distribution was obviously associated with different water masses in the research area. *Paracalanus aculeatus*, which was considered a warm water epipelagic species, was reported as an indicator species of the KC in the research area [48]. In the present study, *P. aculeatus* was mainly obtained in the mixed water area around Kueishan Island. Stations around Keelung Island were controlled by northeast monsoon-derived water masses characterized by the following indicator species: *T. turbinata*, *C. elliptica*, and *C. pauper*. The meso-epipelagic *T. turbinata*, which was the dominant species in the research area, is considered an indicator species of warm water masses and is widely distributed in tropical, subtropical, and temperate areas [35,50,56,82]. Tseng et al. [82] demonstrated that *C. elliptica* significantly and negatively correlated with seawater temperature. Previous research also reported that *C. pauper* is a warm-water species common around the island of Taiwan [82,83]. *Copilia mirabilis* is known for its tropical affinity, being considered a significant indicator species of El Niño in the Gulf of California [84,85]. In the present research, *C. mirabilis* was mainly found in the oceanic area with an IndVal of 92.01%. This high value was supported by previous studies [85,86,87].

In the future ocean, copepod populations will be more strongly affected by warming rather than by acidification. However, ocean acidification effects can modify some temperature impacts, such as ocean acidification and water warming antagonistically will impact the copepod body size and the nutritional composition [88]. Low pH could reduce the reproductive traits of copepods, such as egg size, egg production rate, hatching success, and naupliar production [89,90]. The egg production rate of copepods significantly increases with increased temperature at relatively higher pH values but significantly decreases with increased temperatures at relatively lower pH values [91]. DNA damage could be elevated after long-term exposure to lower pH and then elevates copepod mortality [92]. A negative response was also observed in this research, such as in the dominant species *P. parvus*, *T. turbinata*, and other copepods. Other dominant species did not show a correlation with pH values in this study. During this research, surface waters were mainly dominated by northeast monsoon-derived water masses, Kuroshio intrusion water masses, and mixed water masses in the northeast of Taiwan. These three water masses could be easily distinguished based on temperature, dissolved oxygen, and pH. Bio-indicator species were *T. turbinata*, *C. elliptica*, and *C. pauper* for the northeast monsoon-derived water masses. *Farranula concinna* and *C. mirabilis* were suitable indicators for Kuroshio intrusion water masses in the research area. In the mixed water masses, the indicator species were *P. truncata*, *O. clevei*, and *P. aculeatus*. Calanoid copepodites represented the most abundant group in this study. The horizontal distribution of all indicator species showed a significantly heterogeneous pattern, which was in accordance with water mass distribution in the research area during this research. Apart from calanoid copepodites, the dominant species were *P. aculeatus*, *C. furcatus*, *C. pauper*, *Oncaea venusta*, *Acrocalanus gracilis*, *P. parvus*, *T. turbinata*, *Farranula gibbula*, and *Oncaea media*. Among these species, *P. parvus* and *T. turbinata* were more sensitive than other dominant species to pH value variation in the research area. Surface water temperature and salinity were the main factors that influenced the abundance and distribution of *P. aculeatus*, *C*. *pauper*, *P. parvus,* and *T. turbinata*.

## 5. Conclusions

Our results support the two hypotheses of this study: (1) the interaction between the eastern seawater and the Kuroshio water in the northeastern waters of Taiwan during the period when the northeast monsoon prevailed, resulting in the observed geographic distribution of planktonic copepods; (2) the environmental factors, temperature, and salinity, affected the composition and abundance of the dominant species. The results of the present study demonstrated that the distribution of indicator species could provide a suitable indicator of changes in water masses. The surface water temperature and dissolved oxygen were the main environmental factors affecting the dominant copepod species composition and distribution in the waters of northern Taiwan.

## Figures and Tables

**Figure 1 biology-11-01357-f001:**
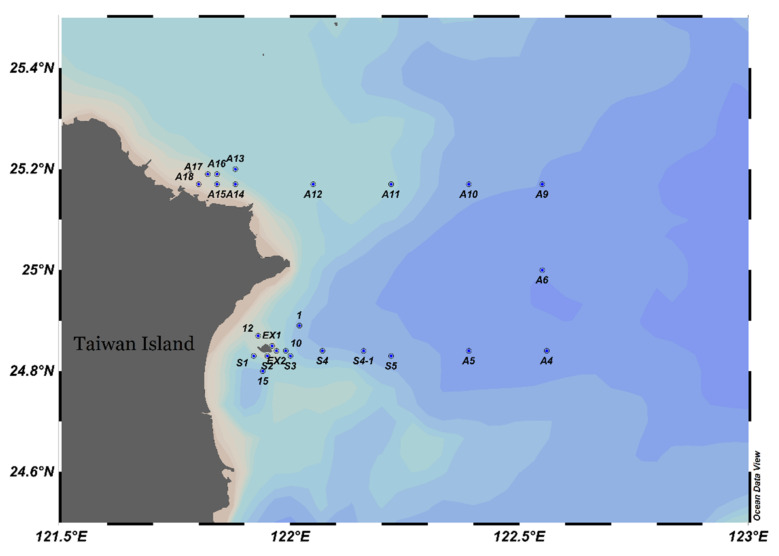
Research area and sampling stations.

**Figure 2 biology-11-01357-f002:**
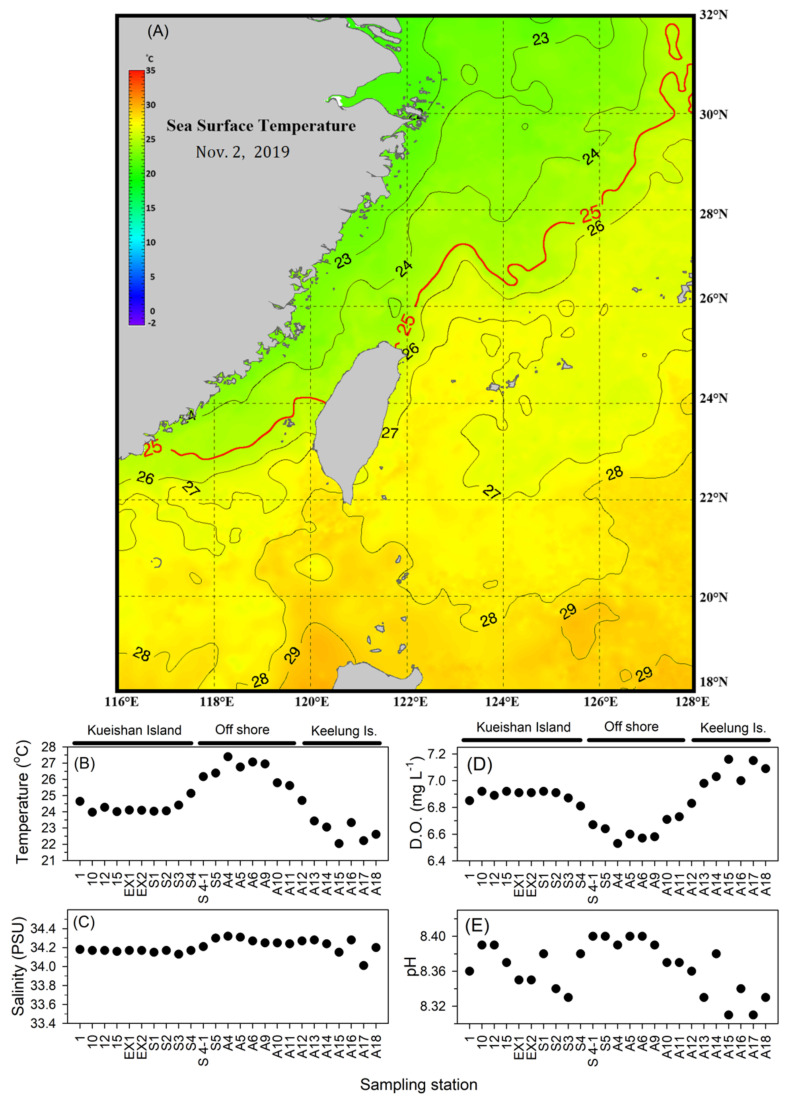
Surface water environmental parameter characteristics during the research. (**A**) Sea surface water temperature changes; (**B**) Temperature; (**C**) Salinity; (**D**) Dissolved oxygen; (**E**) pH.

**Figure 3 biology-11-01357-f003:**
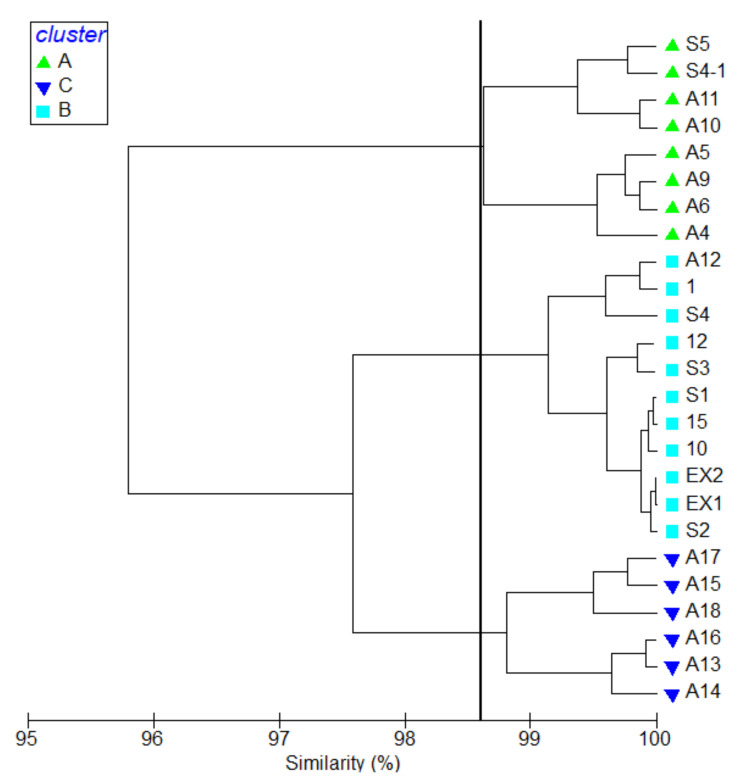
Cluster analysis of stations indicating 3 station groups: A, B, C.

**Figure 4 biology-11-01357-f004:**
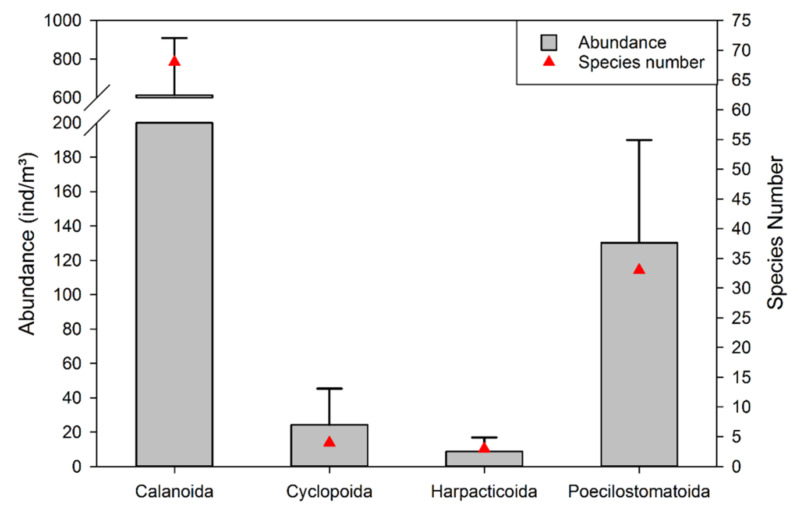
Species number and abundance composition in the research area during autumn.

**Figure 5 biology-11-01357-f005:**
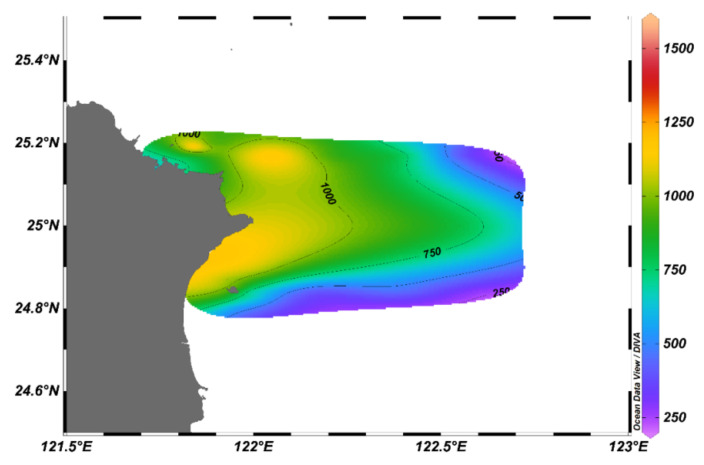
Horizontal distribution of copepod abundances in the study area.

**Figure 6 biology-11-01357-f006:**
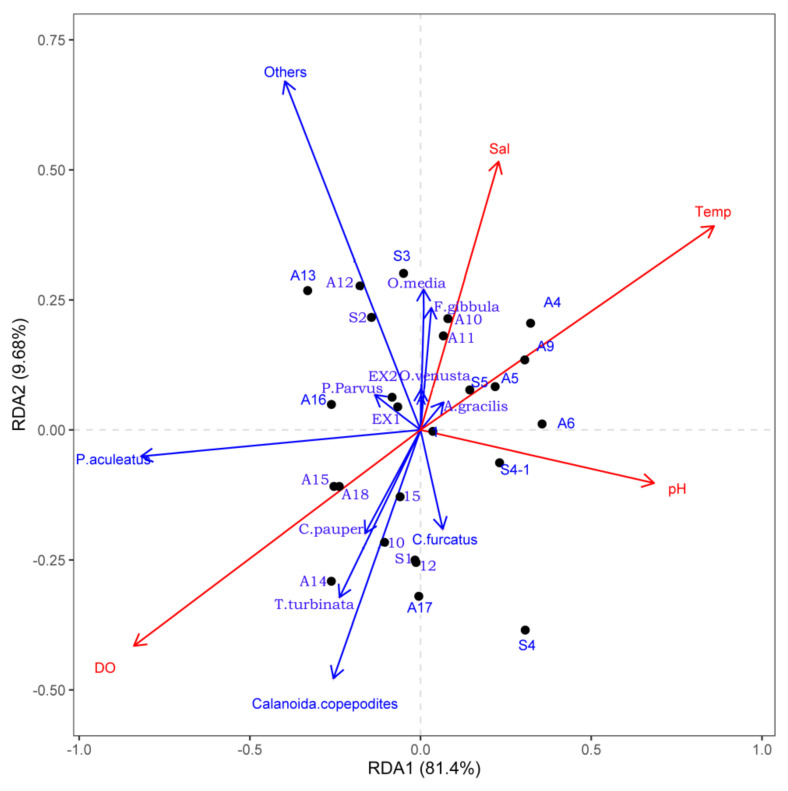
RDA analysis of dominant species and environmental factors in the study area (Temp: temperature; Sal: salinity; DO: dissolved oxygen).

**Table 1 biology-11-01357-t001:** Mean abundance (ind.m^−3^), relative abundance (RA) and occurrence rate (OR) of dominant species in the research area.

Species	MA	RA (%)	OR (%)
*Calanoid copepodites*	217.24	28.06	100.0
*Paracalanus aculeatus*	142.79	18.44	88.0
*Clausocalanus furcatus*	37.14	4.80	96.0
*Canthocalanus pauper*	27.79	3.59	96.0
*Oncaea venusta*	19.30	2.49	100.0
*Acrocalanus gracilis*	20.10	2.60	92.0
*Paracalanus parvus*	25.02	3.23	68.0
*Temora turbinata*	25.89	3.34	64.0
*Farranula gibbula*	18.08	2.34	88.0
*Oncaea media*	18.36	2.37	72.0

**Table 2 biology-11-01357-t002:** Mean abundance (inds. m^−3^) and IndVal (%) of indicator species in different cluster groups (Figure 3).

Species	Group A		Group B		Group C	
	Abundance	IndVal	Abundance	IndVal	Abundance	IndVal
*Acartia pacifica*	**18.83**	**58.83**	7.8	23.92	0.04	0.04
*Acrocalanus monachus*	1.47	3.86	0.63	1.8	**4.25**	**50.22**
*Calanopia elliptica*	**8.43**	**67.25**	1.71	11.91	0.31	0.73
*Candacia aethiopica*	0.17	1.29	0.35	5.88	**1.66**	**66.60**
*Candacia* (copepodites)	/	/	/	/	**5.15**	**75.00**
*Canthocalanus pauper*	**60.86**	**65.25**	23.44	22.85	8.96	9.61
*Centropages calaninus*	/	/	0.25	4.89	**1.13**	**61.56**
*Centropages furcatus*	0.27	14.72	**1.28**	**52.4**	/	/
*Copilia mirabilis*	0.1	1.49	0.17	2.74	**3.09**	**92.01**
*Corycaeus agilis*	4.03	2.1	10.26	23.38	**17.63**	**55.22**
*Corycaeus speciosus*	**7.62**	**55.44**	1.52	12.07	2.31	15.15
*Euchaeta concinna*	**12.29**	**61.12**	4.36	23.68	0.1	0.15
*Euchaeta* (copepodites)	31.1	20.96	**62.2**	**51.45**	5.61	1.42
*Farranula concinna*	1.31	0.62	0.69	0.71	**33.32**	**94.36**
*Farranula gibbula*	8.34	10.67	17.4	30.38	**26.33**	**50.58**
*Macrosetella gracilis*	**15.41**	**57.31**	7.08	26.35	4.39	16.34
*Oncaea clevei*	8.74	18.86	**21.31**	**55.19**	8.56	11.09
*Paracalanus aculeatus*	184.75	44.96	**217.13**	**52.83**	9.09	1.38
*Paracalanus parvus*	**42.82**	**53.58**	23.94	19.06	13.16	8.23
*Paracandacia truncata*	0.17	8.03	**1.18**	**69.47**	0.04	1.53
*Temora discaudata*	**10.66**	**61.82**	4.5	26.08	2.09	7.56
*Temora turbinata*	**78.28**	**69.06**	16.07	15.47	0.1	0.01

## Data Availability

Not applicable.

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
