# Peer review of "Copepods as Indicators of Different Water Masses during the Northeast Monsoon Prevailing Period in the Northeast Taiwan"

_biology, 2022, doi:10.3390/biology11091357_

Round 1

Reviewer 1 Report

Thank you for the scrupulous work with plankton species identification and counting! Precise taxonomic knowledge of occurring species is of a great value for global assessments of biodiversity in the seas and oceans.

Still, there are several issues with the manuscript to be corrected, the most important ones are listed below:

1) The overall purpose of the paper is not clear and the research question or hypothesis does not appear anywhere. What is the novelty of this research? What is the unknown to be revealed, considering the vast amount of references cited?

2) One-time survey of a few days like in your study provides opportunity to assess the seasonal composition of species and the distribution of species in relation to water masses at the respective moment. In no way such survey can be used to estimate any ecological (causal) relationship between copepods and the environment! Monitoring of at least several seasons is needed to judge upon these relationships. Thus, the correlations calculated in the manuscript does not provide reliable information and, for example, statements on species sensitivity to pH values in the area are just speculations. 

3) the clustering exercise of the stations mostly demonstrates presence of statistical skills, as the distinguishing oceanographic features of three sampled areas are clearly obvious from the plotted values in Fig.2 (A-D). Besides, Fig. 2E needs more explanation in the text of the manuscript - the reference curves themselves do not provide much information. 

4) performance of ANOVA for comparison of abundance between groups A, B and C of the pelagic copepods also does not seem very helpful, as the differences are so well expressed in the absolute numbers. 

5) principles for both the description of results on the indicator species as well as discussion on these species are not quite comprehensible. In the results more harmonised approach to describe the indicator species is needed for all groups (lines 243-248).  The discussion on occurrence of indicator species (lines 325-335) does not provide any additional explanation - three mentioned indicator species of water masses around Keelung Island have different preferences for temperature, yet they all are find together. Why so? Then, the dominant species of the studied area (Table 1) do not include all indicator species, yet it is not discussed anywhere. 

6) please have an extensive language check for the style and grammar in the manuscript. 

Author Response

Reviewer 1

  1. The overall purpose of the paper is not clear and the research question or hypothesis does not appear anywhere. What is the novelty of this research? What is the unknown to be revealed, considering the vast amount of references cited?

Answer: Manuscript was modified as the reviewer suggested.

  1. One-time survey of a few days like in your study provides opportunity to assess the seasonal composition of species and the distribution of species in relation to water masses at the respective moment. In no way such survey can be used to estimate any ecological (causal) relationship between copepods and the environment! Monitoring of at least several seasons is needed to judge upon these relationships. Thus, the correlations calculated in the manuscript do not provide reliable information and, for example, statements on species sensitivity to pH values in the area are just speculations.

Answer: The title was changed based on content and restrictions of our study.

  1. The clustering exercise of the stations mostly demonstrates presence of statistical skills, as the distinguishing oceanographic features of three sampled areas are clearly obvious from the plotted values in Fig.2 (A-D). Besides, Fig. 2E needs more explanation in the text of the manuscript - the reference curves themselves do not provide much information.

Answer: Figure 2 was replaced by a new one and the TS-diagram was removed. A SST figure was added to demonstrate the interaction of different water masses in the research area.

  1. Performance of ANOVA for comparison of abundance between groups A, B and C of the pelagic copepods also does not seem very helpful, as the differences are so well expressed in the absolute numbers.

Answer: As the RESULTS demonstrated, the research stations were well clustering into three groups based on the environmental parameters but not according to the species composition and abundance. One-way ANOVA analysis was done to demonstrate the difference of species composition and abundance between different ecological groups.

  1. Principles for both the description of results on the indicator species as well as discussion on these species are not quite comprehensible. In the results more harmonized approach to describe the indicator species is needed for all groups (lines 243-248). The discussion on occurrence of indicator species (lines 325-335) does not provide any additional explanation - three mentioned indicator species of water masses around Keelung Island have different preferences for temperature, yet they all are find together. Why so? Then, the dominant species of the studied area (Table 1) do not include all indicator species, yet it is not discussed anywhere.

Answer: Previously studies have proven the warm-water ecological characteristics of Temora turbinata, Calanopia elliptica, and Canthocalanus pauper. During our research, due to the cold water intrusion which was caused by the northeast monsoon, the surface water temperature was relative lower than that at Kusishan Island and the oceanic area. Our environmental research results also proved a moderate water temperature around Keelung Island. So these three species were found and to be indicator species for Keelung Island.

There was no obviously correlation between the dominant species and indicator species. The dominant species was calculated based on considering as the whole of study area. However, the indicator species was calculated based on the different ecological community. If all the indicator species were included in the dominant species, there will be no indicator meaning for different ecological community.

  1. Please have an extensive language check for the style and grammar in the manuscript.

Answer: This was done by American Englisch Professor Samuel Denny (Seoul).

Reviewer 2 Report

The authors studied the distribution of the planktonic copepods in the Northeast waters of Taiwan in relation of surface water properties.

Based on the environmental parameters, stations were arranged into three groups attributed to different water masses; and indicator species were determined for each group. This study contributes to our knowledge about planktonic ecology of the region and thus is of interest for regional marine ecology. However, the results are flawed by incorrect statistical analysis and poorly presented.

Methods

The method of cluster-analysis is insufficiently described. Which similarity measure has been used? Were the values of environmental variables standardized before the analysis? Which clustering algorithm was applied?

Furthermore, the ANOVA tests for differences should be only applied to groups of samples specified prior to collecting the data. The case if one uses a cluster analysis to define sample groupings and then establishes the statistical validity by performing a test for differences among those groups, is entirely wrong and dangerous misconception, because of the completely circular reasoning (see, e.g., Clarke et al. (2014) Change in marine communities: An Approach to Statistical Analysis and Interpretation). Instead, the reliability and integrity of groups could be approved by post hoc permutational tests (SIMPROF or bootstrapping procedures).

Correlation analysis: Pair-wise Pearson correlations were used to identify significant relationships in case of “taxa-factors” comparison. This “each-to-each” individual testing approach is fully unacceptable and can produce many false-significant correlations, because of the problem of multiple comparisons (see, e.g., Glantz “Primer of Biostatistics”, 1994; or Quinn & Keough “Experimental design and data analysis for biologists”, 2002). Correction for the multiple comparisons should be done when estimating the statistical significance (p-values). I recommend the method proposed by Benjamini and Yekutieli (The Annals of Statistics 2001, 29(4):1165–1188), which controls the false discovery rate in case of inter-correlated environmental variables. The sequential Bonferroni-Holm procedure, which is computationally simple, controls over Type I error but provides enough power for individual tests, is also acceptable solution.

Moreover, instead of (or in addition to) a single species-based approach, it would be desirable to apply a kind of multivariate analysis, to reveal the relationship between copepod community structure and environmental drivers. Such contemporary and widely used methods as Canonical Correspondence Analysis (CCA) or distance-based linear modelling (DistLM) are obviously spring to mind here.

RESULTS

- On the TS-diagram (Fig. 2E), stations showed more or less continuous trend of surface temperature and situated strictly between the Taiwan Strait and Kuroshio Current TS-lines, and therefore could not be attributed to any of these water masses. Furthermore, the first TS-diagram represents 60 m depth water layer, the second (and more cold-water) one – 200-m layer, while the depth range from which the sampling data were obtained, is not indicated directly. If, as I can suppose, it was 5-meter surface layer, these three datasets are not comparable.

Presentation of the results needs to be improved.

- Fig. 1: Color scale on the map filling should be explained (I suppose it indicates the depth?). I would also like to see the main eddies and currents on this map – it would help the readers to conceive easier the hydrological background of the study.

- Spatial distribution of environmental parameters (Fig 2A) is poorly presented. The X-Y graph with arbitrarily arranged stations is not informative; map with variable values for the stations indicated by symbol size or (preferably) by color (like at Fig. 5) would be more suitable here.

- Similarly, the color-scaled maps visualizing the distribution of three groups of indicator species would be desirable.

- L. 203-206: “…Cyclopoida and Harpacticoida accounted for a relatively low percentage of about 3.70% and 2.78%, respectively. The average abundance of Cyclopoida and Harpacticoida were 24.31±20.98 and 8.70±8.15 (inds.m-3), respectively; with a total of 4.01% of the total average abundance altogether”. Please be precise in figures: 3.70% + 2.78% = 6.48% (not 4.01)!

So, I recommend Major revision, with special attention to statistical methods and improving the graphical presentation of the data.

Author Response

Reviewer 2

  1. The method of cluster-analysis is insufficiently described. Which similarity measure has been used? Were the values of environmental variables standardized before the analysis? Which clustering algorithm was applied?

Answer: The manuscript was modified as the referee suggested. The data transformation method and the clustering algorithm were added in the manuscript.

  1. Furthermore, the ANOVA tests for differences should be only applied to groups of samples specified prior to collecting the data. The case if one uses a cluster analysis to define sample groupings and then establishes the statistical validity by performing a test for differences among those groups, is entirely wrong and dangerous misconception, because of the completely circular reasoning (see, e.g., Clarke et al. (2014) Change in marine communities: An Approach to Statistical Analysis and Interpretation). Instead, the reliability and integrity of groups could be approved by post hoc permutational tests (SIMPROF or bootstrapping procedures).

Answer: Following above suggestion stations were grouped into different groups based on environmental parameters. The simprof test was used to evaluate the similarity significance levels.

  1. Correlation analysis: Pair-wise Pearson correlations were used to identify significant relationships in case of “taxa-factors” comparison. This “each-to-each” individual testing approach is fully unacceptable and can produce many false-significant correlations, because of the problem of multiple comparisons (see, e.g., Glantz “Primer of Biostatistics”, 1994; or Quinn & Keough “Experimental design and data analysis for biologists”, 2002). Correction for the multiple comparisons should be done when estimating the statistical significance (p-values). I recommend the method proposed by Benjamini and Yekutieli (The Annals of Statistics 2001, 29(4):1165–1188), which controls the false discovery rate in case of inter-correlated environmental variables. The sequential Bonferroni-Holm procedure, which is computationally simple, controls over Type I error but provides enough power for individual tests, is also acceptable solution.

Answer: Following the reviewer’s suggestion, the Pearson correlation analysis was removed from the manuscript. The relationship of dominant copepod data and environmental factors was analyzed by RDA analysis in the revised version of the MS.

  1. Moreover, instead of (or in addition to) a single species-based approach, it would be desirable to apply a kind of multivariate analysis, to reveal the relationship between copepod community structure and environmental drivers. Such contemporary and widely used methods as Canonical Correspondence Analysis (CCA) or distance-based linear modelling (DistLM) are obviously spring to mind here.

Answer: Accordingly was the RDA analysis replacing the former Pearson correlation to demonstrate the relationship between dominant copepod and environmental factors.

  1. On the TS-diagram (Fig. 2E), stations showed more or less continuous trend of surface temperature and situated strictly between the Taiwan Strait and Kuroshio Current TS-lines, and therefore could not be attributed to any of these water masses. Furthermore, the first TS-diagram represents 60 m depth water layer, the second (and more cold-water) one – 200-m layer, while the depth range from which the sampling data were obtained, is not indicated directly. If, as I can suppose, it was 5-meter surface layer, these three datasets are not comparable.

Answer: Figure 2 was replaced by a new one and the TS-diagram was removed. A SST figure was added to demonstrate the different water mass interactions in the research area.

  1. Presentation of the results needs to be improved.

Answer: Given the new analytical approaches and exchange of figures the RESULTS were improved and the new aspects got discussed.

  1. Fig. 1: Color scale on the map filling should be explained (I suppose it indicates the depth?). I would also like to see the main eddies and currents on this map – it would help the readers to conceive easier the hydrological background of the study.

Answer: Accordingly, a SST figure was added to the manuscript in order to demonstrate the main eddies and currents indicated by the isopleths of the sea surface water temperature.

  1. Spatial distribution of environmental parameters (Fig 2A) is poorly presented. The X-Y graph with arbitrarily arranged stations is not informative; map with variable values for the stations indicated by symbol size or (preferably) by color (like at Fig. 5) would be more suitable here.

Answer: In the revised version in figure 2, the stations were arranged according to the cluster result. It may be more readable together with the sampling station map (figure 1) and the assemblage figure (figure 3) of environmental parameter. The color distribution maps were also  supplied in the appendix.

  1. Similarly, color-scaled maps visualizing the distribution of three groups of indicator species would be desirable.

Answer: The revosed manuscript was modified as the reviewer suggested. The color-scaled maps of the indicator species distribution patterns were submitted as an appendix.

  1. L. 203-206: “…Cyclopoida and Harpacticoida accounted for a relatively low percentage of about 3.70% and 2.78%, respectively. The average abundance of Cyclopoida and Harpacticoida were 24.31±20.98 and 8.70±8.15 (inds.m-3), respectively; with a total of 4.01% of the total average abundance altogether”. Please be precise in figures: 3.70% + 2.78% = 6.48% (not 4.01)!

Answer: In the revised version, the percentage of species number and abundance were calculated and modified carefully.

  1. So, I recommend a Major Revision, with special attention to statistical methods and improving the graphical presentation of the data.

Answer: Thank you for your valuable comments. We have revised and improved the revised manuscript according to your comments.

Round 2

Reviewer 1 Report

Thank you for the corrections! I have no more comments.

Reviewer 2 Report

All my previous comments have been taken in account properly in the amended version of this paper. Methods of data analysis have been improved, and all flaws in the text have been revised. I have no further comments, and suggest this contribution is now suitable for publication.